# Prevalence and predictors of depressive symptoms among school-going adolescents in Kenya

Lucy Magige *, Zipporah Bukania, Moses Mwangi, Vyolah Chuchu, Violet Wanjihia, Sarah Karanja, Stephen Onteri, Antony Macharia, Linnet Ongeri

Kenya Medical Research Institute, Nairobi, Kenya

* lmagige@kemri.go.ke

## Abstract

Depression is a growing concern among adolescents, particularly in low- and middle-income countries (LMICs), such as Kenya, where mental health issues often remain unaddressed. This study assessed the prevalence and predictors of probable depression among adolescents in selected public secondary schools in rural Kenya during the post-COVID-19 era.This cross-sectional study was conducted among adolescents aged 13–19 years selected from thirty (30) public secondary schools in rural Kenya via a two-stage cluster sampling method. The calculated sample was distributed proportionally across schools by type, gender, and grade to ensure representativeness. Written informed consent and assent were obtained from all the respondents. The Patient Health Questionnaire for Adolescents (PHQ-A) was used to screen for probable depression and a cutoff of ≥ 10 was adopted. The CRAFFT (Car, Relax, Alone, Forget, Friends, and Trouble) and Childhood Adolescent Trauma Screener (CATS) tools screened for substance use and trauma exposure, respectively. The study had 1,020 participants (467 males,553 females) with a median age of 17 years (range 13–19 years) and a response rate of 98.3%. Overall prevalence for probable depression was 52.6% based on the PHQ-A cut-off of ≥ 10. Approximately 5% (47) of the participants were at high risk for substance use-related problems, whereas 17.2% (173) exhibited probable posttraumatic stress disorder (PTSD). Age, living away from one's parents, school related pressure, conflicts at home, trauma exposure, and a history of substance use were strongly linked to probable depression. Conversely, living with parents, holding religious beliefs, having accessed guidance and counselling services and having a trusted adult to confide in were associated with a reduced risk for probable depression. Cumulatively, prevalence of probable depression was significantly high among the participants and strongly correlated with psychosocial factors underscoring the importance of integrating routine mental health screening into school health programs.

**Data availability statement:** Data used to generate the findings shared in this publication are hosted at the Kenya Medical Research Institute (KEMRI) under the custodianship of the project's Principal Investigator and Kenya Medical Research Institute the Scientific Ethics and Review Unit (KEMRI-SERU). Due to the ethical approval conditions restricting sharing publicly participant raw data, de-identified data may be made available upon reasonable request for the purposes of reproducibility or statistical verification. Request to access the data can be made to the Principal Investigator, Lucy Magige at lmagige@kemri.go.ke or through the KEMRI-SERU Kenya Medical Research Institute Scientific and Ethics Review Unit (SERU) at seru@kemri.go.ke.

**Funding:** This study was funded by the Government of Kenya through the Kenya Medical Research Institute(KEMRI) Internal Research Grants (KEMRI/IRG/EC007 to L.M.). No authors received a salary from this funder. The funders had no role in study design, data collection and analysis, decision to publish, or preparation of the manuscript.

**Competing interests:** The authors have declared that no competing interests exist.

**Abbreviations:** LMICs: Low-and- middle income countries; PHQ-A: Patient Health Questionnaire for Adolescents; CRAFFT: Car, Relax, Alone, Forget, Friends, and Trouble; CATS: Childhood Adolescent Trauma Screener; PTSD: posttraumatic stress disorder; PTSS: post traumatic stress symptoms; DALYs: Disability Adjusted Life Years; SERU: Scientific and Ethics Review Unit; NACOSTI: National Commission for Science, Technology & Innovation; MoE: Ministry of Education; MoH: Ministry of Health.

## Introduction

Adolescence represents a critical developmental phase and a high-risk window for the onset of depressive symptoms, which manifest as feelings of loneliness, sadness, loss of interest, displeasure in activities, and disrupted regulatory functions [1,2].

Globally, depression is among the leading causes of illness and disability among adolescents, making it a public health concern [3]. A systematic review and meta-analysis revealed a concerningly high global point prevalence rate of elevated self-reported depressive symptoms, estimated at 34% between 2001 and 2020 [4]. While a literature review by Carrelas et al. established a point prevalence of 29.2% for subclinical depressive symptoms among adolescents under the age of 18 years [5].

Notably, the prevalence has been on an increasing trajectory with the prevalence of depressive symptoms increasing from 24% in the period of 2001–2010 to 37% between 2011 and 2020 [4]. These consistent surge in prevalence rates is alarming [1], since depressive symptoms increase dramatically during adolescence as a result of rapid physiological, psychological, and sociocultural changes [3]. Research has shown that by the age of 19 years, approximately 25% of adolescents experience a depressive episode [6], and approximately 70% of adolescents report recurrent depressive symptoms within 3–5 years after the initial episode [7]. Unfortunately, half of all cases of depressive symptoms remain undetected and untreated until later in life resulting in serious negative immediate and long-term impacts on the individual's health and non-health outcomes [8]. These effects can vary from low academic achievement, poor sleep quality, and increased substance use to more severe outcomes such as self-harm or suicide [6,9]

The global burden of depressive symptoms among adolescents is disproportionate with low-income and middle-income countries (LMICs) and sub-Saharan Africa (SSA) bearing the highest burden. An umbrella review of mental health disorders among adolescents in Africa established a pooled prevalence of 37.8% for depression, a figure that surpasses global estimates and further reinforces the evident crisis in the region [10]. Similarly, another systematic review revealed 26.9% median point prevalence of depression across nine studies of the general adolescent population [11]. However, a separate meta-analysis for SSA reported a lower pooled prevalence rate of 15.27% for depression [12]. This difference between the figures reported demonstrates the fragmented and heterogenous nature of the research landscape in the SSA, calling for more evidence to show the true burden.

Similar to the SSA, Kenya broadly exemplifies a typical scenario of the wider challenges and complexities of adolescent mental health. Although at national level data remains scarce, a few local studies have shown evidence of the high burden of particularly depressive symptoms among school going adolescents that the country bears. According to the Group Health Mental Adolescent Kenya, in 2019, depressive symptoms accounted for the most (3.1%) DALYs of mental disorders among 10–24-year-olds in Kenya [13]. A study conducted among school-going adolescents from Nairobi and the coastal region of Kenya reported a 20.6% prevalence of

depressive symptoms [14], whereas Mokaya et al. reported a 36.4% prevalence of depressive symptoms among high school students in Nairobi [15]. However, another study conducted in Makueni County, Kenya which is considered rural reported a striking prevalence rate of 58.9% [16], showing that adolescents from rural and semi-rural areas could potentially be severely affected by this growing pandemic [16].

The adolescents are uniquely sensitive to psychosocial cues meaning that depressive symptoms during this stage is multifactorial, and influenced by a multitude of interconnected factors. Individual factors such age, gender, socioeconomic and environmental factors such as financial hardships, adverse childhood trauma, history of suicide attempts within family, unstable living conditions and family dynamics, academic stress and access to mental health services have been reported to be significant predictors for depressive symptoms [10,17–20]. On the other hand, some of the potential protective factors that have been reported to mitigate the risk of depressive symptoms among this population include social support, family cohesion, having strong religious beliefs and access to mental health care [21–23].

Adolescents are particularly vulnerable to adverse effects such as pandemics and desire more social interactions, meaning that disruptions such like the COVID-19 pandemic exacerbated the already worrying high rates of depressive symptoms [24–26]. The pandemic and the associated preventive public health measures such as social distancing, prolonged school closure and economic disruptions, quickly became a new stressor impacting the health and emotional well-being of adolescents [27,28]. For context, the school environment provides a foundational pillar for routine and social network and as such, this disruption had a direct negative impact on the students and could have been worse especially in the rural and semi-rural areas due to other socio-economic challenges. According to a global meta-analysis conducted on 29 studies, the pooled prevalence estimates of clinically elevated child and adolescent depressive symptoms doubled from 12.9% before the pandemic to 25.2% after the pandemic [2]. A similar systematic review and meta-analysis on the extent of mental health problems among adolescents (13–18 years) during the COVID-19 pandemic established a pooled prevalence of 34% (95% CI: 18.2%, 50.7%; $p < 0.001$) [29]. A study conducted in Kenya on the impact of the COVID-19 pandemic on mental health indicated that a high proportion of adolescents reported experiencing depressive symptoms during the pandemic due to containment measures [25,30].

Despite the high burden and serious consequences of mental health problems, particularly among adolescents, the subject remains under prioritized, especially in sub-Saharan countries such as Kenya. Moreover, existing studies in the region have not systematically investigated the influence of childhood trauma and substance use on the outcomes of depression among adolescents. More specifically, limited research has been conducted in public secondary schools, especially those in rural areas, which are more prone to socio-economic shocks [31]. Kakamega County was purposively selected since it is the most populous rural county in Kenya and faces severe health challenges, such as high rates of drug and substance abuse and teenage pregnancy among young people, leading to high school drop outs [32]. This has been attributed to poverty, food insecurity, low access to health care services and peer pressure.

Most countries have developed policies to promote mental health, and some countries, such as Kenya, have further developed mental health action plans (2021–2025), which emphasize the need to curate targeted school mental health programmes to promote mental health among learners [33]. However, additional data on the burden of mental health is needed to guide these initiatives.

Therefore, These findings are both timely, relevant and provide critical insights into the burden and predictors of depressive symptoms in adolescents with local and broad implications to guide the development of such programmes.

## Methodology

### Study objective

To examine prevalence and predictors for probable depression among adolescents post the COVID-19 pandemic in Kakamega County, Kenya.

PLOS Mental Health

## Study design

This research employed a cross-sectional study design with a concurrent mixed methods approach. However, the findings presented in this paper are limited to quantitative data and qualitative findings will be published separately.

## Study population

Adolescents aged 13–19 years attending public secondary schools.

## Study area

This study was conducted in Kakamega County, which is located in the western part of Kenya and is approximately 400 km from the capital city of Nairobi. As the second most populous county after Nairobi with the largest rural population [34],the county faces multidimensional poverty (poor housing conditions, poor nutrition, poor sanitation facilities, low access to education and low levels of economic activities) [35].

Specifically, the study was carried out in public secondary schools located in the sub-counties of Butere, Malava and Matungu. These sub-counties were selected to capture the county's socio-economic diversity with Butere and Malava sub-counties representing semi-rural areas and Matungu sub-county, capturing the rural parts of the county [34]. Fig 1 below represents the geographical location of the county and distribution of the sub-counties.

## Sample size determination

The study sample size was calculated using Fisher's formula for proportion-type outcomes with 5% precision with 95% confidence interval [36]. Considering the prevalence of depressive symptoms among school-going mid-adolescents in Kenya reported at 43.7% and applying a design effect of 2.5 to account for clustering at the school level, the required sample size was estimated to be 945 participants. To account for potential non-response and missing data, an additional 10% was added, yielding a final rounded target sample size of 1,050 students. This sample size was considered sufficient to provide precise estimates of probable depression within narrow confidence interval.

## Sampling

A total of 30 schools were sampled from the three sub-counties. A two-stage cluster sampling approach, facilitated by a sampling frame generated from an updated school enrolment list collected 1–2 weeks before the survey was used to distribute the sample size as detailed in Table 1 below;

## Participants selection

Participants were recruited between January and March 2023. Adolescents who attended public, day, or boarding secondary schools; >18 years of age who consented and <18yrs who assented following a parental consent were eligible for inclusion. Participants who exhibited acute, intrusive levels of psychiatric symptoms and did not provide consent or assent, including those whose parents did not provide consent, were excluded.

For Potential participants who were in boarding schools, consent forms were issued to the students during the school half term break for the parents/guardians to give consent, thereafter the study team would follow with a telephone call to explain the study details and take them through the consent form. For those who could not be reached via phone or lived near the school premises, the parents were asked to come to school after the mid-term break and provide consent. If the participant was completely unavailable, a replacement was done using random sampling while maintaining the same criteria of proportionate to gender and class allocation.

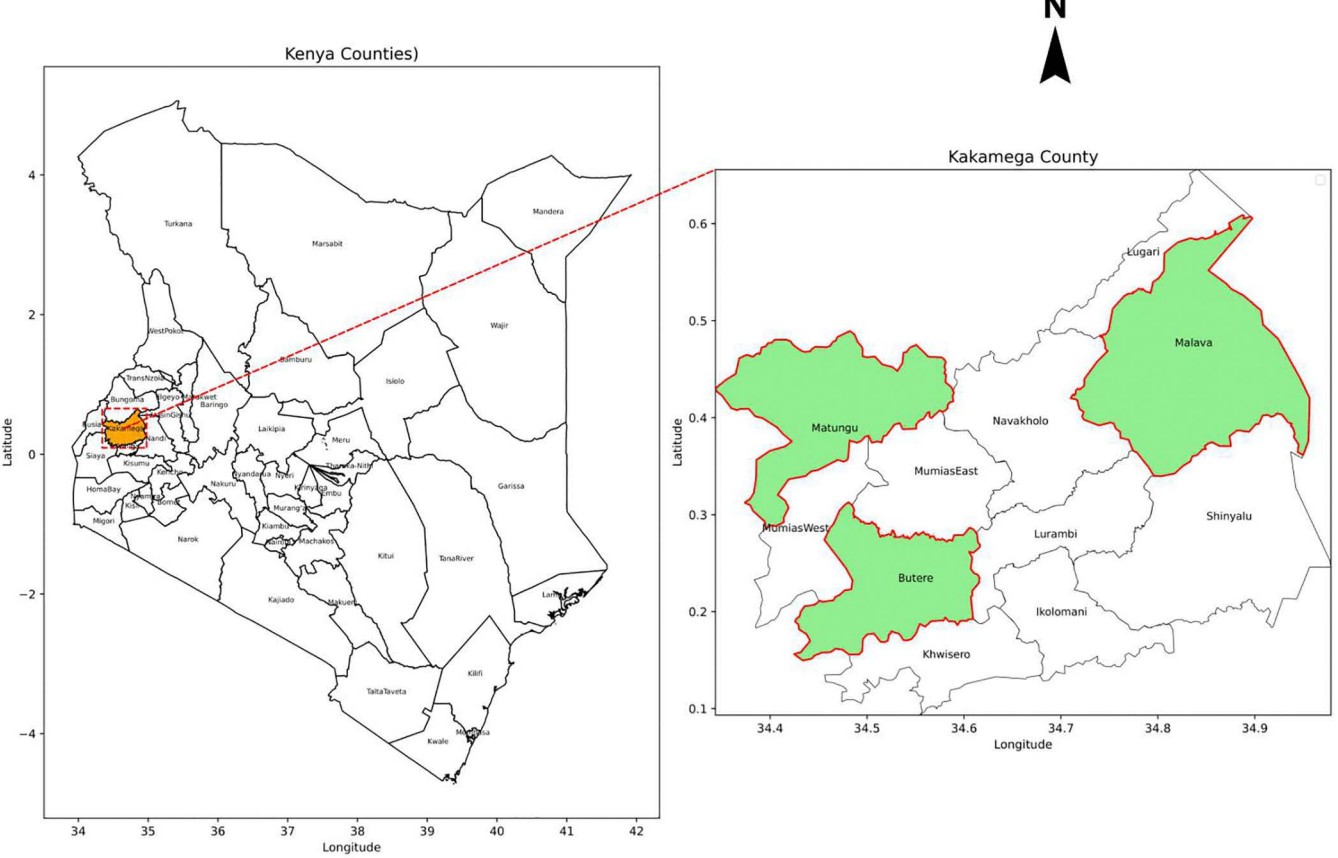

Base layer of map; https://gadm.org/maps/KEN.html

License; https://gadm.org/license.html

**Fig 1. Map showing the three study sub-counties within Kakamega County.** https://gadm.org/maps/KEN.html.

**Table 1. Distribution of sample size across the sub-counties, school type and gender.**

| Sub-County | School-type | | | Gender | | | Achieved population | Calculated sample size | Total Student population |
|---|---|---|---|---|---|---|---|---|---|
| | Boys only | Girls only | | | | | Mixed school (boys & girls) | Male | Female |
| Butere | 1 | 3 | 5 | 101 | 191 | | 292 | 299 | 3120 |
| Malava | 3 | 1 | 6 | 188 | 169 | | 357 | 352 | 3709 |
| Matungu | 1 | 1 | 9 | 179 | 192 | | 371 | 387 | 3278 |
| **Total** | **5** | **5** | **20** | **468** | **552** | | **1,020** | **1,038 ≈ 1,050** | **10,107** |

## Ethics statement

Ethical clearance was received from the Kenya Medical Research Institute Scientific and Ethics Review Unit (KEMRI-SERU) (41-03-2022/4500), research permits from the National Commission for Science, Technology & Innovation (NACO-STI) (NACOSTI/P/22/20792**),** and authorization from the Kenya Ministry of Education (MoE) and the Ministry of Health (MoH). Permission was obtained from the Kakamega County director of education and the school management board and administration. All participants provided informed written assent, and written informed consent was obtained from their parents or guardians.

## Data collection

Data was collected during the first term of the academic calendar year, just after the mid-term break (mid to end-March).

Prior to data collection, research assistants who were undergraduate students studying psychology and from the study county were recruited to support field work activities. Apart from efficiency in fieldwork and cost management in terms of logistics, recruiting locally was critical for cultural competency and contextual understanding of the social dynamics and cultural norms. Additionally, a background in psychology, helped them understand and follow the study protocol in less time and they were able recognize distress cues, respond and refer appropriately. This helped improve the overall reliability and validity of findings as they were able to probe and minimize bias by applying standard interview skills.

The research assistants were trained for a week on the study protocol, building rapport with the participants to ensure a smooth interview process and how to ask questions in a way that ensures cultural sensitivity. After which, they had to take part in a supervised pretest and piloting exercise and based on what was observed, concepts were clarified and errors rectified. This process also ensured cultural validation of the PHQ-A in addition to the fact that it had also been validated in Kenya for use among adolescents [37].

To safeguard the well-being of the participants, at the end of each interview, a list of all students who had requested for counselling services or had been identified as distressed was handed over to the guidance and counselling teacher with notes on whether the student was comfortable confiding in them or an alternative person, in this case a representative from the mental health services department at the County referral hospital whom we had informed prior to the study initiation.

**Variables**: The dependent variable was probable depression, as the study utilized a screening tool, and the independent variables were risk factors such as psychosocial factors, childhood and adolescent trauma and substance abuse.

## Data sources/measurements

The Patient Health Questionnaire modified for Adolescents (PHQ-A), was used to screen for probable depression and was researcher administered. The tool is a 9-item depression screening tool and the overall scoring was weighted based on individual question score. Each question had 4 options (Not at all = 0 Several Days = 1 More than half the days = 2 Nearly every day = 3). A score of 0–4 was suggestive of none or minimal depression, 5–9 mild depression, 10–14 moderate depression, 15–19 moderately severe depression and 20–27 severe depressive symptoms. A cutoff score of ≥10 on the PHQ-A was used to define probable depression. The PHQ-A is a validated tool that has been used in adolescent populations in South Africa [38].

To examine substance, use as a risk factor for probable depression, the CRAFFT test questionnaire was used. Its name is a mnemonic of the first letters of key words in the test's 6 questions and stands for Car, Relax, Alone, Forget, Friends and Trouble. The CRAFFT was research administered, and each yes answer = 1 point. This questionnaire was developed specifically for use among adolescents. It was validated in Zambia and found to have good reliability [39].

Additionally, the Childhood Adolescent Trauma Screener (CATS) which is a short-validated trauma screening instrument was administered to measure potentially traumatic events and posttraumatic stress symptoms (PTSSs) a risk factor

for probable depression. The CATS has 15 items measuring traumatic events, 20 items measuring DSM-5 PTSD symptoms, and 5 items measuring psychosocial functioning. A total symptom score was calculated by summing up the raw scores of items 1–20 (possible range = 0–60). An international validation study demonstrated good to excellent reliability, with α values ranging between 0.88 and 0.94 [40].

Other risk factors, such as family history of suicidal behaviour, bullying, school stress and family structure, which have been cited in previous studies, were assessed via a researcher-designed questionnaire. content validity, a mental health expert who is part of the investigators reviewed it and provided initial feedback to revise the tool. We also pretested and piloted the questionnaire in two schools separate from the sampled schools but within the study sub-counties. Feedback from these two activities further guided the refinement of the items to ensure clarity, cultural appropriateness, internal consistency and reliability. The refined tool was then administered in the main survey.

## Statistical analysis plan

Descriptive statistics included all socio demographic characteristics, support factors and psychological factors and are presented as counts and percentages for categorical variables and as measures of central tendency for continuous variables, i.e., means (SDs) for normally distributed variables and medians (IQRs) for skewed distributions.

The analysis of socio-demographic characteristics, support factors and psychological factors associated with probable depression was performed via two control strategies. To identify factors associated with probable depression (before adjustment for confounding), a crude estimation of the associations between probable depression and each independent factor was conducted via Pearson's chi-square test or Fisher's exact test (where expected cell counts were <5). The Mann–Whitney U test was used to compare the differences in continuous independent variables between participants with probable depression and those without. Crude odds ratios (OR) with corresponding 95% confidence intervals (CI) were used to measure the strength of the associations.

To identify factors independently associated with probable depression and to control for potential confounding, a more liberal threshold ($p < 0.10$) was applied during the bivariate screening stage to avoid excluding predictors that might become significant after adjustment. In addition, key theoretically plausible variables, such as age and sex, were retained in the multivariable analysis regardless of their bivariate significance. Two complementary modeling approaches were applied to identify independent predictors of probable depression. First, an exploratory stepwise binary logistic regression (forward and backward conditional) was performed. This procedure retained the following predictors: *Age, living with parents, Living with others, Lost member to suicide, Trusted adult, Conflict at home, Pressure at school and CATS scores*. Second, a penalized regression model using least absolute shrinkage and selection operator (LASSO) was fitted to address potential multicollinearity, reduce overfitting and improve predictor stability. The optimal penalty parameter (λ) was chosen via 10-fold cross-validation. Predictors with non-zero coefficients were retained and these variables were refitted in a standard logistic regression model to obtain adjusted odds ratios, 95% confidence intervals and p-values. LASSO selected a slightly different but more parsimonious set of predictors: *Age, living with parents, Lost member to suicide, accessed guidance and counseling, Conflict at home, Pressure at school, CATS scores.* Given its stronger performance and robustness, the LASSO-selected set of predictors was used to construct the final multivariable logistic regression model. Importantly, *Gender*, although not retained in either selection method, was forced back into the final model due to its theoretical and epidemiological plausibility.

Model assumptions and diagnostics were carefully evaluated. Tolerance values for all predictors were >0.5 and variance inflation factors (VIFs) were <2, indicating absence of multicollinearity. The model was statistically significant on the Omnibus test of model coefficients ($\chi^2 = 199.5$, df = 10, $p < 0.001$). The model explained a moderate proportion of variance in probable depression (Cox & Snell R² = 0.201; Nagelkerke R² = 0.342). Calibration was adequate, as shown by a non-significant Hosmer–Lemeshow test ($\chi^2 = 9.91$, df = 8, p = 0.272), exemplifying that predicted

probabilities did not significantly differ from observed outcomes. The model correctly classified 86.7% of the participants with a specificity of 95.7% and a sensitivity of 40.3%. ROC analysis shown good discriminatory ability (AUC = 0.84; 95% CI: 0.81–0.87, p < 0.001).

The threshold for statistical significance was set at p < 0.05 (2-tailed) for all inferential statistics. All statistical analyses were performed using SPSS version 27.0 and R version 4.5.1 with RStudio version 2025.9.0.387 as the integrated development environment.

To address missing data, since there were small amounts of item-level missing data (i.e., 1013 completed CRAFFT, 1006 completed CATS), these cases were excluded listwise for the specific analyses especially in the Multivariable analysis. Given the small proportion of missingness (<2%) and most being derived variables as the ones mentioned, no imputation was applied.

## Sensitivity analysis

To evaluate the robustness of the final model, several sensitivity analyses were undertaken. First, alternative PHQ-A cutoffs (≥ 5, ≥ 8, ≥ 10, and ≥ 15) were applied to further evaluate probable depression. Across these thresholds, the direction and magnitude of associations for the key predictors remained largely consistent except on the higher cutoff ≥15 where most of the predictors were insignificant. Second, potential interaction effects were also examined, i.e. *age × gender*, but no statistically significant interactions were observed. Third, alternative predicted probability thresholds were assessed. The conventional cutoff of 0.50 yielded high specificity but lower sensitivity, whereas ROC analysis identified an optimal probability cutoff of 0.09, which substantially improved sensitivity while retaining moderate specificity as evident in Table 2.

We also conducted the Bootstrap resampling (1,000 iterations) to assess further the predictors stability. The strongest predictors were *CATS scores* [Probable PTSD] (100%), *Accessed Guidance* and counselling (93.2%), *Pressure at School* (90.0%), *Conflict at Home* (87.1%), *CATS scores [Moderate trauma-related distress]* (86.3%) and Lost *member to suicide* (81.0%), all of which were retained in more than 80.0% of replications. Other predictors such as *Age* (75.8%), *living with parents* (71.1%), *Trusted Adult* (64.1%) and *Gender* (25.5%) were retained at different proportions of the bootstrap samples as highlighted.

**Table 2. Sensitivity analysis across different PHQ-A cutoffs and probability thresholds.**

| Cutoff/ Adjustment | Sensitivity (%) | Specificity (%) | AUC | Significant Predictors* |
|---|---|---|---|---|
| PHQ-A ≥ 5 | 70.5 | 77.0 | 0.83 | Age, Lost member to suicide, Conflict at home, Pressure at school, CATS scores. |
| PHQ-A ≥ 8 | 47.1 | 91.0 | 0.83 | Age, Gender, living with parents, Lost member to suicide, Conflict at home, Pressure at school, CATS scores. |
| PHQ-A ≥ 10 (main definition) | 40.3 | 95.7 | 0.84 | Age, Lost member to suicide, Conflict at home, Pressure at school, CATS scores and Accessed guidance and counselling. |
| PHQ-A ≥ 15 | 3.2 | 99.8 | 0.83 | Lost member to suicide and CATS scores |
| Predicted prob ≥ 0.50 | 40.3 | 95.7 | 0.84 | Age, Lost member to suicide, Conflict at home, Pressure at school, CATS scores, and Accessed guidance and counselling. |
| Predicted prob ≥ 0.30 | 56.9 | 90.1 | 0.84 | Age, Lost member to suicide, Conflict at home, Pressure at school, CATS scores, and Accessed guidance and counselling. |
| Predicted prob ≥ 0.09 (Optimal by Youden's J) | 86.1 | 64.2 | 0.84 | Age, Lost member to suicide, Conflict at home, Pressure at school, CATS scores, and Accessed guidance and counselling. |

Note: * The predictors of the final multivariable logistic regression model (PHQ-A ≥ 10) found to be sig (p < 0.05) after adjustment of the PHQ-A cutoff or Predicted probability.

## Results

### Response rate

A total of 1,020 respondents out of a target sample of 1,050 participants participated in the study. The response rate was 98.3%.

### Socio-demographic characteristics

Majority of the respondents (69.30%) attended sub-county-level secondary schools, with the largest proportion enrolled in mixed (boys and girls) schools (69.30%), and over 80.0% attending day schools. The median age was 17 years with an age range of 13–19 years. It is important to mention that in rural Kenyan secondary schools, it is typical to find students who are older than 18 years old due to repeated grades or having to drop out of school because of lack of school fees or pregnancy, of which some rejoin later when their age has progressed. In regards to gender, female students constituted more than half (553) of the total respondents. With respect to class distribution, students in the 1st year and 3rd year of secondary schools(equivalent to Grades 9–11; ages 14–17 years) had nearly equal representation, whereas the 4th year secondary school students (Grade 12; ages 17–18 years) accounted for the smallest proportion (19.4%), showing the reduced rate of progression to higher grades for most students in the rural areas which could be attributed to various socio-economic challenges. For religious affiliation, over 90.0% of the respondents identified as Christians. Although 88.3% reported the availability of counselling services in school, only 36.6% had utilized them. When asked about the source of funding for their school fees, most reported predominantly being supported by parents or family members (Table 3).

### Depressive symptoms and risk factors

Among the 1,020 adolescents screened for probable depression, 2.9% exhibited signs of moderately severe depression, 12.4% moderate depression and 36.9% mild depression based on the PHQ-A cut-off; a score of 0–4 was suggestive of none or minimal depression, 5–14 suggests mild to moderate depression. 15–19 moderate to severe depression and 20–27 suggests severe depressive symptoms Fig 2. Among the 1,013 adolescents screened, approximately 5.0% were at high risk of substance abuse-related problems. The remaining adolescents had a score of less than 2, indicating medium to low risk Fig 3. Among the 1,006 adolescents screened for possible signs of childhood adolescent trauma, 17.2% exhibited probable posttraumatic stress disorder (PTSD), and 15.8% reported moderate trauma-related distress. Over 60.0% of the participants were within the normal ranges, i.e., not clinically elevated symptoms of trauma Fig 4. Values used to plot can be found in S1 Data.

### Socio-demographic characteristics associated with probable depression

The results of the bivariate analysis of socio-demographic characteristics presented in Table 4 indicate that probable depression was significantly associated with sub-county of residence, age, level of class and religion. Students who resided in Matungu sub-county, which was selected to represent the rural population, had significantly greater odds of depression (18.8%) than their counterparts in Butere sub-county (11.6%), which represented the semi-rural population (OR = 1.76, p = 0.012). The likelihood of probable depression also increased with each additional year of age by 31.0% and it was still significant even after adjusting for other variables, such as sex, class, religion, sub-county, school category and school type (aOR = 1.23, p = 0.009).

Although the rate of probable depression was slightly higher in female adolescents (16.6%) than in their male counterparts, the difference was not statistically significant (OR = 1.15, p = 0.417) even after adjusting for other variables (aOR = 1.36, p = 0.167). The bivariate analysis also revealed that probable depression rates increased with each successive grade. First year secondary students reported the lowest rate (9.8%) of probable depression although this proportion

**Table 3. Socio-demographic characteristics of study participants(N = 1,020).**

| Characteristics | n (%) | Median (Range/IQR) |
|---|---|---|
| **School category** | | |
| County | 313(30.7) | |
| Sub-county | 707(69.3) | |
| **School type** | | |
| Boys only school | 138(13.5) | |
| Girls only school | 175(17.2) | |
| Mixed School | 707(69.3) | |
| **Age (Years)** | | 17.0 (13–19); IQR = 2 (16–18) |
| **Gender** | | |
| Male | 467(45.8) | |
| Female | 553(54.2) | |
| **Class level** | | |
| 1st year of secondary school | 287(28.1) | |
| 2nd year of secondary school | 245(24.0) | |
| 3rd year of secondary school | 290(28.4) | |
| 4th year of secondary school | 198(19.4) | |
| **Living status*** | | |
| Parents | 730(71.6) | |
| Alone | 6(0.6) | |
| Relatives | 272(26.7) | |
| Friends | 5(0.5) | |
| School | 31(3.0) | |
| Others | 24(2.4) | |
| **Religion** | | |
| Christian | 929(91.1) | |
| Islam | 86(8.4) | |
| Non-religious | 5(0.5) | |
| **School attendance type** | | |
| Day | 856(83.9) | |
| Boarding | 164(16.1) | |
| **Availability of guidance and counselling** | | |
| Yes | 901(88.3) | |
| No | 30(2.9) | |
| Don't know | 89(8.7) | |
| **Accessed guidance and counselling†** | (n = 901) | |
| Yes | 374(36.6) | |
| **Mode of school fees funding*** | | |
| Government | 102(10.0) | |
| Family, parent | 974(95.5) | |
| Harambee(fundraising) | 1(0.1) | |
| Church, companies, NGO | 49(4.8) | |

Notes: *Multiple response variable; †Among those reporting availability of guidance and counselling (n = 901).

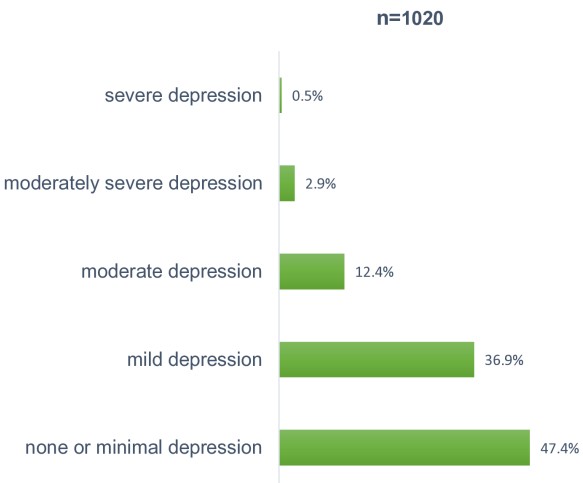

Fig 2. **Prevalence of depressive symptoms.**

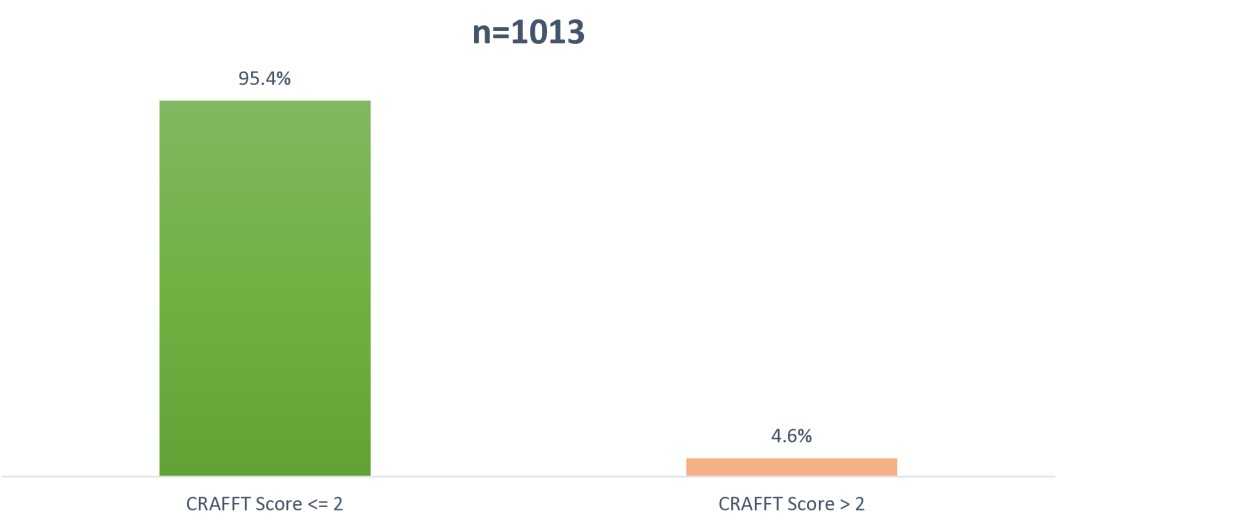

Fig 3. **Prevalence of substance use disorder.**

doubles in the third(OR = 2.31, p = 0.001) and fourth (OR = 2.42, p = 0.001) year of secondary school. Compared to participants who reported being religious, individuals who were non-religious had a markedly greater proportion of probable depression (60.0%) and significantly greater odds (OR = 7.92, p = 0.024).

Multivariable analysis, adjusting for socio-demographic characteristics, support and psychological factors, revealed that probable depression was significantly associated with increasing age (aOR=1.23; 95% CI = 1.05--1.44; p = 0.009). However, the associations between religion and grade level were not statistically significant.

## Home support structures associated with probable depression

Living environments and family structure significantly influence the likelihood of depression as depicted by the odds of probable depression being 1.91 times greater for those who did not live with their parents than for those who did (OR = 1.91,

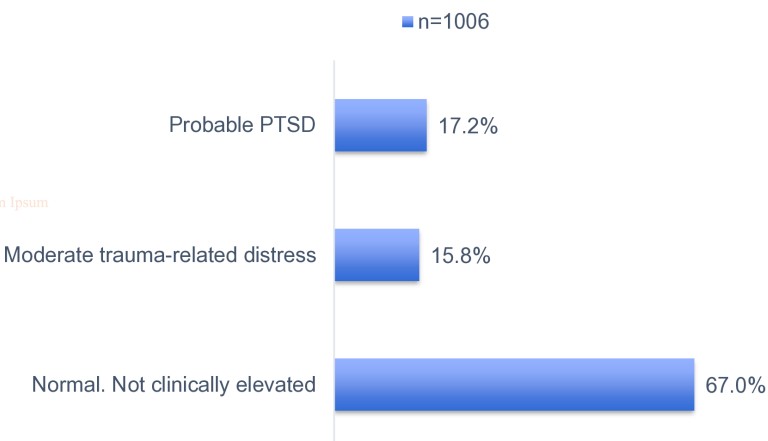

**Fig 4. Prevalence of probable posttraumatic stress disorder (PTSD).**

**Table 4. Socio-demographic characteristics associated with probable depression.**

| Variable | Probable Depression n(%) | No depression n(%) | OR (95% CI) | p value | aOR (95% CI) | p -value |
|---|---|---|---|---|---|---|
| **Sub county** | | | | | | |
| Butere | 33(11.6) | 251(88.4) | Ref | | | |
| Malava | 55(15.8) | 293(84.2) | 1.43(0.90-2.27) | 0.132 | | |
| Matungu | 73(18.8) | 315(81.2) | 1.76(1.13-2.75) | **0.012** | | |
| **Category of school** | | | | | | |
| County | 42(13.4) | 271(86.6) | Ref | | | |
| Subcounty | 119(16.8) | 588(83.2) | 1.31(0.89-1.91) | 0.169 | | |
| **Age in years** | | | 1.31(1.15-1.48) | **<0.001** | 1.23(1.05-1.44) | **0.009** |
| **Type of school** | | | | | | |
| Boys only school | 19(13.8) | 119(86.2) | Ref | | | |
| Girls only school | 27(15.4) | 148(84.6) | 1.14(0.61-2.16) | 0.681 | | |
| Mixed School | 115(16.3) | 592(83.7) | 1.22(0.72-2.05) | 0.463 | | |
| **Sex** | | | | | | |
| Male | 69(14.8) | 398(85.2) | Ref | | Ref | |
| Female | 92(16.6) | 461(83.4) | 1.15(0.82-1.62) | 0.417 | 1.35(0.88-2.08) | 0.175 |
| **Grade level** | | | | | | |
| 1st year of secondary school | 28(9.8) | 259(90.2) | Ref | | | |
| 2nd year of secondary school | 34(13.9) | 211(86.1) | 1.49(0.88-2.54) | 0.142 | | |
| 3rd year of secondary school | 58(20.0) | 232(80.0) | 2.31(1.42-3.75) | **0.001** | | |
| 4th year of secondary school | 41(20.7) | 157(79.3) | 2.42(1.44-4.06) | **0.001** | | |
| **Religion** | | | | | | |
| Christian | 148(15.9) | 781(84.1) | Ref | | | |
| Islam | 10(11.6) | 76(88.4) | 0.69(0.35-1.37) | 0.300 | | |
| Non-religious | 3(60.0) | 2(40.0) | 7.92(1.31-47.78) | **0.024** | | |

**Note:** Ref = Reference category; OR = Odds ratio; aOR = Adjusted odds ratio; CI = Confidence interval.

95% CI: 1.24–2.86, p = 0.002). Furthermore, the odds of probable depression were significantly higher (OR = 3.33, 95% CI: 1.43--7.75, p = 0.005) for those living with non-family members.

Multivariable analysis revealed that, after adjusting for socio-demographic characteristics, home/school support structures and psychological factors, probable depression was insignificantly associated with not living with parents (aOR=1.54; 95% CI = 1.54(0.99--2.35; p = 0.055). However,, there was nosignificant association established between probable depression and other living arrangements, such as living alone, with relatives, friends, at school, or having different family structures (nuclear, single parent, or extended family).

Although not statistically significant, adolescents from nuclear families with two parents had lower odds of probable depression (OR = 0.67, 95% CI: 0.29--1.58, p = 0.359) compared to those from single parents and extended family structures, who had over 20.0% risk for probable depression. This was highlighted by the bivariate analysis of home support structures presented in Table 5 below.

## School support structures associated with probable depression

The bivariate analysis of the school support structures presented in Table 6 indicated that most of the participants with high odds of probable depression were those who reported to having their school fees not paid by family or parents compared to individuals whose family members covered their school fees (OR = 2.46, 95% CI: 1.28-4.73, p = 0.007).

**Table 5. Home support structures associated with probable depression.**

| Variable | Probable Depression n (%) | No depression n (%) | OR (95% CI) | p value | aOR (95% CI) | p value |
|---|---|---|---|---|---|---|
| **Living alone** | | | | | | |
| Yes | 2(33.3) | 4(66.7) | 2.69(0.49-14.8) | 0.256 | | |
| No | 159(15.7) | 855(84.3) | Ref | | | |
| **Living with parents** | | | | | | |
| Yes | 100(13.7) | 630(86.3) | Ref | | Ref | |
| No | 61(21.0) | 229(79.0) | 1.91(1.24-2.86) | **0.002** | 1.54(0.99-2.38) | 0.055 |
| **Living with relatives** | | | | | | |
| Yes | 45(16.5) | 227(83.5) | 1.08(0.74-1.57) | 0.688 | | |
| No | 116(15.5) | 632(84.5) | Ref | | | |
| **Living with friends** | | | | | | |
| Yes | 0(0.0) | 5(100.0) | ND | ND | | |
| No | 161(15.9) | 854(84.1) | Ref | | | |
| **Living in school** | | | | | | |
| Yes | 5(16.1) | 26(83.9) | 1.03(0.39-2.71) | 0.957 | | |
| No | 156(15.8) | 833(84.2) | Ref | | | |
| **Living with others** | | | | | | |
| Yes | 9(37.5) | 15(62.5) | 3.33(1.43-7.75) | **0.005** | | |
| No | 152(15.3) | 844(84.7) | Ref | | | |
| **Family structure** | | | | | | |
| Nuclear: two parents | 72(12.8) | 491(87.2) | 0.67(0.29-1.58) | 0.359 | | |
| Nuclear: Single parent | 31(21.1) | 116(78.9) | 1.22(0.49-3.03) | 0.666 | | |
| Extended family | 36(21.6) | 131(78.4) | 1.26(0.51-3.08) | 0.618 | | |
| Two parent home | 15(15.0) | 85(85.0) | 0.81(0.3-2.16) | 0.669 | | |
| Others | 7(17.9) | 32(82.1) | Ref | | | |

**Note:** Ref = Reference category; ND = non-definitive due to zero cases; OR = Odds ratio; aOR = Adjusted odds ratio; CI = Confidence interval.

**Table 6. School support structures associated with probable depression.**

| Variable | Probable Depression n (%) | No depression n (%) | OR (95% CI) | p value | aOR (95% CI) | p value |
|---|---|---|---|---|---|---|
| **School attendance type** | | | | | | |
| Boarding | 133(15.5) | 723(84.5) | Ref | | | |
| Day | 28(17.1) | 136(82.9) | 1.12(0.72-1.75) | 0.621 | | |
| **School fees Government** | | | | | | |
| Yes | 11(10.8) | 91(89.2) | Ref | | | |
| No | 150(16.3) | 768(83.7) | 1.62(0.84-3.10) | 0.148 | | |
| **School fees family, parent** | | | | | | |
| Yes | 147(15.1) | 827(84.9) | Ref | | | |
| No | 14(30.4) | 32(69.6) | 2.46(1.28-4.73) | **0.007** | | |
| **School fees churches, companies, NGOs** | | | | | | |
| Yes | 12(24.5) | 37(75.5) | 1.79(0.91-3.51) | 0.091 | | |
| No | 149(15.3) | 822(84.7) | Ref | | | |
| **Availability of guidance and counselling** | | | | 0.096 | | |
| Don't know | 9(10.1) | 80(89.9) | 0.59(0.29-1.21) | 0.148 | | |
| No | 8(26.7) | 22(73.3) | 1.91(0.84-4.38) | 0.125 | | |
| Yes | 144(16.0) | 757(84.0) | Ref | | | |
| **Ever accessed guidance and counselling** | | | | | | |
| No | 108(18.9) | 463(81.1) | 1.9(1.27-2.86) | **0.002** | 2.36(1.46-3.81) | **<0.001** |
| Yes | 36(10.9) | 294(89.1) | Ref | | Ref | |

**Note:** Ref = Reference category; OR = Odds ratio; aOR = Adjusted odds ratio; CI = Confidence interval.

Multivariable analysis revealed that students who had never accessed guidance and counselling services had significantly greater odds of probable depression both before (OR = 1.90, p = 0.002) and after adjusting for socio-demographic characteristics, home/school support structures and psychological factors (OR=2.36; 95% CI = 1.46--3.81; p < 0.001) than those who had accessed guidance and counselling.

It is important to note, that students attending day secondary schools presented a slightly higher prevalence of probable depression (17.1%) compared to those who were in boarding schools (15.5%).

## Psychological factors associated with probable depression

The findings highlight the role of different psychological factors, including trauma exposure, substance abuse, lack of social support, and academic pressure on probable depression as shown in Table 7. At the bivariate analysis level, participants who did not know if they had ever lost a close family member by suicide, did not have a trusted adult to talk to about personal thoughts and stressors, reported a form of conflict at home that was hard to handle, had experienced pressure at school and bullied at school, were at moderate to high risk, or experienced moderate or probable distress.

The data further revealed that students who had lost a close family member by suicide had 1.83 times greater odds of depression than those who had not (p = 0.001). Conversely, students who were unsure if they had lost a close family member by suicide ("Don't know") had significantly greater odds of probable depression both before (OR = 4.62, p < 0.001) and after adjusting (aOR=5.26; 95% CI = 1.72-16; p = 0.004). Having a trusted adult to talk to was a protective factor, as those who did not, had significantly higher odds (1.87 times) of probable depression than those who did (p < 0.001).

**Table 7. Psychological factors associated with probable depression.**

| Variable | Probable Depression n (%) | No Depression n (%) | OR (95% CI) | p value | aOR (95% CI) | p value |
|---|---|---|---|---|---|---|
| **Ever lost a close family member by suicide** | | | | **<0.001** | | **0.011** |
| No | 95(13.0) | 634(87.0) | Ref | | Ref | |
| Yes | 57(21.5) | 208(78.5) | 1.83(1.27-2.63) | **0.001** | 1.29(0.81-2.03) | 0.282 |
| Don't know | 9(40.9) | 13(59.1) | 4.62(1.92-11.10) | **<0.001** | 5.26(1.72-16.12) | **0.004** |
| **Having a trusted adult to talk to about personal thoughts and stressors** | | | | | | |
| No | 57(22.6) | 195(77.4) | 1.87(1.30-2.68) | **<0.001** | | |
| Yes | 104(13.5) | 664(86.5) | Ref | | | |
| **Ever seen a therapist/counsellor** | | | | | | |
| No | 118(14.8) | 682(85.3) | Ref | | | |
| Yes | 43(19.6) | 176(80.4) | 1.41(0.96-2.08) | 0.080 | | |
| **Any conflict at home hard to handle** | | | | | | |
| No | 48(9.0) | 486(91.0) | Ref | | Ref | |
| Yes | 113(23.3) | 373(76.7) | 3.07(2.13-4.41) | **<0.001** | 1.67(1.06-2.62) | **0.026** |
| **Ever experienced pressure at school** | | | | | | |
| No | 69(10.1) | 614(89.9) | Ref | | Ref | |
| Yes | 92(27.6) | 241(72.4) | 3.40(2.40-4.80) | **<0.001** | 1.69(1.10-2.61) | **0.017** |
| **Currently bullied at school** | | | | | | |
| No | 123(14.1) | 748(85.9) | Ref | | | |
| Yes | 37(25.3) | 109(74.7) | 2.06(1.36-3.14) | **0.001** | | |
| **Substance abuse (CRAFFT Score)** | | | | | | |
| Low risk (≤2) | 145(15.0) | 821(85.0) | Ref | | | |
| Moderate to high risk (>2) | 16(34.0) | 31(66.0) | 2.92(1.56-5.48) | **0.001** | | |
| **Trauma history (CATS score)** | | | | | | **<0.001** |
| Not clinically elevated (0–14) | 44(6.5) | 630(93.5) | Ref | | Ref | |
| Moderate distress (15–20) | 27(17.0) | 132(83.0) | 2.93(1.75-4.90) | **<0.001** | 2.54(1.44-4.50) | **0.001** |
| Probable PTSD (21–60) | 90(52.0) | 83(48.0) | 15.53(10.13-23.8) | **<0.001** | 10.26(6.24-16.88) | **<0.001** |

**Note:** Ref = Reference category; OR = Odds ratio; aOR = Adjusted odds ratio; CI = Confidence interval.

Furthermore, adolescents who had experienced conflicts at home that were difficult to handle had significantly greater odds of probable depression both before (OR = 3.07, p < 0.001) and after adjustment (aOR=1.67; 95% CI = 1.06--2.62; p = 0.026). A similar trend was also observed among those adolescents who reported to be* experiencing pressure at school, as they had significantly higher odds of depression both before (OR = 3.40, p < 0.001) and after adjustment (aOR = 1.69, 95% CI = 1.10-2.61; p = 0.017). Bullying in school was also significantly associated (p = 0.001) with probable depression, as students who reported being currently bullied were 2.06 times at risk.

Substance abuse was also significantly associated (p = 0.001) with probable depression, as students at moderate to high risk of substance abuse had significantly higher odds (2.92 times) of probable depression than those at low risk. Further, a history of exposure to trauma had the one of the greatest impact on probable depression. The number of adolescents who reported moderate distress on the basis of the CATs score were 2.93 before and 2.54 times after adjustment at risk for probable depression, whereas those who had probable PTSD had extremely high odds of probable depression both before (OR = 15.53, p < 0.001) and after adjustment (aOR = 10.26, p < 0.001).

## Discussion

This study provides critical insights into the high burden of probable depression among adolescents and sheds light on the complex interplay of diverse psychosocial factors that predispose them to mental health conditions.

This study reported an overall prevalence of 52.6% for probable depression, findings that align with other recent studies which have reported similarly elevated prevalence rates among adolescents and higher than global estimates (34.0%) [4]. Further, evidence indicates that the prevalence has been increasing as highlighted by Shorey et al., who reported an increase from 24.0% in 2001–2010 to 37.0% in the subsequent decade(2011–2020) [4].

These results mirror a broader growing trend of mental health issues reported in sub-Saharan Africa, with figures exceeding global estimates as shown in a recent umbrella review that found a pooled depression prevalence of 37.8% for adolescents across the African continent [10]. This picture is further nuanced by additional cross-sectional studies that reported prevalence rates ranging from 29.0-33.9%, varying by local contexts, gender and risk exposures [41–43].These regional variabilities could be a result of methodological heterogeneity or the fragmented research in the region.

This crisis is more severe in Kenya, where recent studies conducted among high school students in similar rural and semi-rural settings reported prevalence rates of 36.4%and 45.9% [16,44,45]. These figures exceed those of older studies conducted in other urban areas of Kenya among similar cohorts that reported prevalence rates in the range of 25.7-29.0% [46,47]. These contextual differences may be a result of methodological disparities or growing stressors. Notably, during the Covid-19 pandemic the prevalence rates of depressive symptoms ranged from 2.5 to 63.8% which is double the global estimates owing to the mandatory containment measures such as school closure and social isolation factors now recognized as significant predictors of depressive symptoms [2,25,26,48,49].

Besides biological and genetic predispositions, psychosocial factors are significant influencers of depressive symptoms during adolescence, affecting both the onset and severity of depression [50]. These factors represent a complex interplay between an individual's psychological development and their social environment operating across individual, familial, and social levels and interacting with demographic and cultural variables.

This study revealed that the likelihood of probable depression increased with age and being in a higher-class level. These findings are consistent with other studies conducted in similar settings that demonstrated that a greater proportion of older adolescent students reported moderate to severe probable depression depressive symptoms [42] and that the odds of experiencing probable depression and anxiety increased with age [14].

Similarly, school-related pressures emerged as a significant risk factor, findings that mirror other studies that have shown a strong linkage between academic stress and adolescent probable depression [51]. The competitive nature of the education system, where end of secondary school examination results greatly influence future academic and career prospects, may explain why school-related stress plays such a significant role in adolescent mental well-being. Many students experience long study hours, frequent assessments, and high parental or teacher expectations, leading to anxiety and burnout [52]. Furthermore, the fear of academic failure, coupled with societal stigma surrounding poor performance, may contribute to feelings of inadequacy and low self-esteem, thereby exacerbating the risk of depression, especially among students in highigher class levels [33].

Additionally, living away from parents and family conflicts were reported as key drivers of mental health well-being, highlighting the role of exposure to negative familial interactions and household stressors on the onset of mental health conditions in the long run [20,45,52,53]. Living away from parents was associated with a higher risk of probable depression, suggesting that social support from parents/caregivers is a significant predictor of probable depression [42]. Furthermore, increased vulnerability could be due to reduced parental monitoring and weaker social support systems [54]. Other studies have demonstrated that inability to cope with family disputes and economic difficulties were found to contribute to a high prevalence of probable depression among adolescents [55,56]. Persistent family discord can result in parental separation or domestic violence and inter-generational disagreements, creating an environment of insecurity, where adolescents may feel emotionally neglected or caught in parental disputes, leading to heightened feelings of anxiety and hopelessness.

Childhood trauma was also strongly associated with probable depression, findings that further reinforce extensive literature linking exposure to early-life traumatic events to long-term psychological distress [20]. These findings align with those of [57,58] who reported that early-life stress increased the likelihood of developing major depressive disorder and exposure to traumatic events increased the risk of probable depression and suicidal behaviours [59]. Adolescents who have a history of trauma, such as abuse, neglect, or the loss of a caregiver, may develop mal-adaptive coping mechanisms, increasing their risk of probable depression [60].

Conversely, this study identified strong protective factors that can mitigate against the risk of probable depression. Being religious, living with parents and having a trusted adult to talk to were significantly associated with lower odds of probable depression. Additionally, adolescents who had ever accessed guidance and counselling services had significantly lower odds for probable depression, findings that further reinforce the need for formal mental health support structures in school settings [61]. These findings corroborate other similar studies that highlight the role of family relationships and social support in reducing vulnerability to mental health conditions [23,42,62]

## Study limitations

The cross-sectional study design of this study highlights associations but does not establish causality, making it unclear whether depression preceded or resulted from associated factors. Additionally, the population group was adolescents attending selected public secondary schools in three sub-counties in a rural county in Kenya, which may limit the generalizability of the findings to other geographic or socio-economic contexts in the country. Furthermore, the study relied on self-reported screening tools which could have suffered from recall and social desirability biases. Despite these limitations, the study has several notable strengths. The systematic and randomized sampling approach minimized biases commonly observed in similar studies. The inclusion of key variables such as day and boarding schools provided a more comprehensive understanding of contextual differences, which are often overlooked in prior research. Moreover, the use of standardized tools allows for a detailed examination of substance use, trauma history, and protective factors such as social support and access to mental health resources. These findings offer valuable insights on probable depression as a public health issue and offers critical insights for policy development and targeted intervention planning.

## Conclusion

This study provides evidence on the burden of depressive symptoms among school-going adolescents in Kenya and highlights the interplay between multifaceted risk factors such associo-demographic factors, family conflicts, school-related pressures, and childhood trauma that play significant roles in the high incidence of depression.

As schools present the ideal set-up for intervention entry, the findings emphasize the need for urgent targeted interventions that address both individual and structural determinants of adolescent mental health. Implementing or enhancing structured school-based mental health programs such as regular screening protocols, stress management workshops, and academic counselling services may help mitigate the negative impact of poor adolescent mental health. Furthermore, there is a need to enhance the capacity of teachers to identify early warning signs and implement management protocols.

To address familial conflicts, there is a need to foster supportive family environments through targeted interventions, such as parental counselling and conflict resolution programs within the community, which can play a critical role in mitigating the adverse effects on the mental health well-being of young persons. Similarly, interventions that provide trauma-informed care, early psychological counselling, and community-based mental health services that are youth friendly could help mitigate the long-term impact of childhood trauma on adolescent probable depression.

Future research could explore longitudinal trends in adolescent probable depression, examine differences in risk factors between private and public secondary schools and evaluate the effectiveness of school- and community-based interventions aimed at promoting mental well-being in this vulnerable population.

## Supporting information

**S1 Data. Dataset used to generate** Figs 2**,** 3 **and** 4 **showing distribution of depressive symptoms (PHQ-A), substance use risk (CRAFFT Score), and trauma-related distress (CATs Scores) among the study population.** (XLSX)

## Author contributions

**Conceptualization:** Lucy Magige, Zipporah Bukania, Moses Mwangi, Sarah Karanja, Linnet Ongeri.

**Data curation:** Lucy Magige, Moses Mwangi, Antony Macharia.

**Formal analysis:** Moses Mwangi, Antony Macharia.

**Funding acquisition:** Lucy Magige.

**Investigation:** Lucy Magige, Vyolah Chuchu, Sarah Karanja, Stephen Onteri.

**Methodology:** Lucy Magige, Zipporah Bukania, Moses Mwangi, Vyolah Chuchu, Violet Wanjihia, Sarah Karanja, Stephen Onteri, Antony Macharia, Linnet Ongeri.

**Project administration:** Lucy Magige, Zipporah Bukania, Vyolah Chuchu, Violet Wanjihia, Linnet Ongeri.

**Supervision:** Zipporah Bukania, Violet Wanjihia, Linnet Ongeri.

**Validation:** Lucy Magige, Zipporah Bukania, Moses Mwangi, Violet Wanjihia, Sarah Karanja, Stephen Onteri, Antony Macharia, Linnet Ongeri.

**Visualization:** Lucy Magige, Moses Mwangi, Violet Wanjihia, Sarah Karanja, Stephen Onteri, Antony Macharia, Linnet Ongeri.

**Writing – original draft:** Lucy Magige, Moses Mwangi, Vyolah Chuchu, Antony Macharia.

**Writing – review & editing:** Zipporah Bukania, Moses Mwangi, Vyolah Chuchu, Violet Wanjihia, Sarah Karanja, Stephen Onteri, Antony Macharia, Linnet Ongeri.

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
