## [Decision Letter · Decision Letter 0]

30 Jul 2025

PMEN-D-25-00266

Prevalence and Predictors of Depressive Symptoms among School-Going Adolescents in Kenya.

PLOS Mental Health

Dear Dr. Magige,

Thank you for submitting your manuscript to PLOS Mental Health. After careful consideration, we feel that it has merit but does not fully meet PLOS Mental Health’s publication criteria as it currently stands. Therefore, we invite you to submit a revised version of the manuscript that addresses the points raised during the review process.

We look forward to receiving your revised manuscript.

Kind regards,

Vitalii Klymchuk, Ph.D., D.Sc.

Academic Editor

PLOS Mental Health

Journal Requirements:

1. We have amended your Competing Interest statement to comply with journal style. We kindly ask that you double check the statement and let us know if anything is incorrect.

2. Your current Financial Disclosure states, “Government of Kenya through KEMRI Internal Research Grants”. However, your funding information on the submission form indicates that you received funding from “Government of the Republic of Kenya” with grant number KEMRI/IRG/EC007 from

Ms Lucy Magige. Please indicate by return email the full and correct funding information for your study and confirm the order in which funding contributions should appear. Please be sure to indicate whether the funders played any role in the study design, data collection and analysis, decision to publish, or preparation of the manuscript.

3. We note that your Data Availability Statement is currently as follows: [All data generated or analysed during this study are included in this published article]

Additional Editor Comments (if provided):

Reviewers' comments:

Reviewer's Responses to Questions

**Comments to the Author**

1. Does this manuscript meet PLOS Mental Health’s publication criteria ? Is the manuscript technically sound, and do the data support the conclusions? The manuscript must describe methodologically and ethically rigorous research with conclusions that are appropriately drawn based on the data presented.

Reviewer #1: Yes

Reviewer #2: Yes

Reviewer #3: Yes

Reviewer #4: Yes

2. Has the statistical analysis been performed appropriately and rigorously?

Reviewer #1: Yes

Reviewer #2: Yes

Reviewer #3: I don't know

Reviewer #4: N/A

3. Have the authors made all data underlying the findings in their manuscript fully available (please refer to the Data Availability Statement at the start of the manuscript PDF file)?

Reviewer #1: Yes

Reviewer #2: Yes

Reviewer #3: Yes

Reviewer #4: No

4. Is the manuscript presented in an intelligible fashion and written in standard English?

Reviewer #1: Yes

Reviewer #2: Yes

Reviewer #3: Yes

Reviewer #4: Yes

5. Review Comments to the Author

Reviewer #1: Dear Authors

Thank you for the opportunity to review this important manuscript exploring the prevalence and predictors of depressive symptoms among school-going adolescents in rural and semi-rural Kenya. The study is both timely and relevant, addressing a significant public health issue with broad implications, particularly in low- and middle-income country (LMIC) contexts. You demonstrate methodological competence and ethical rigor, and the findings offer clear policy implications for mental health integration into school systems.

After a thorough review, I find that the manuscript is of strong potential for publication in PLOS Mental Health, pending minor revisions. Below is a summary of the strengths and areas that require attention.

Strengths

• The study addresses a well-defined research question using appropriate cross-sectional methodology.

• The sample size is adequate, and the sampling approach (two-stage cluster) enhances representativeness.

• Use of validated screening tools (PHQ-A, CRAFFT, CATS) adds to the methodological robustness.

• The multivariate analysis is detailed and appropriately interpreted.

• The manuscript is well-structured and clearly written.

• Ethical procedures were properly followed and well-documented.

• The findings have strong applicability to mental health policy and school-based interventions in LMICs.

Revisions Required

1. Clarify Terminology of “Depression”

The use of “probable depression” throughout the manuscript may be misinterpreted by readers as a clinical diagnosis. As the PHQ-A is a screening, not diagnostic tool, the authors should ensure that this distinction is clear in the abstract, results, and discussion sections.

2. Limitations: Generalizability

The study is limited to three sub-counties in Kakamega County. The discussion section should more explicitly acknowledge the potential limitations of generalizing the results to other geographic or socioeconomic contexts in Kenya.

3. Timing and Contextual Factors

There is limited information on the timing of data collection in relation to academic calendar events or potential psychosocial stressors. A brief mention of this context would help readers assess external influences on depressive symptoms.

4. Manuscript Language

While generally clear, some sections, particularly the discussion contain repetitive phrasing. A light edit for conciseness and clarity would improve readability.

5. Mixed Methods Mention

Although a mixed methods design is noted, this manuscript presents only quantitative results. Consider either omitting the mixed methods mention or indicating that qualitative findings will be published separately.

Recommendation

The manuscript is suitable for publication pending minor revisions as outlined above. The study provides valuable evidence with strong relevance to adolescent mental health services and educational policy in Kenya and other LMICs.

Thank you again for the opportunity to review this manuscript.

Regards,

Dr David Onchonga

Reviewer #2: 1. Does this manuscript meet PLOS Mental Health’s publication criteria?

Response: Yes

The manuscript addresses a critical and timely public health issue - adolescent depressive symptoms in Kenya, through a cross-sectional study conducted in a resource-limited and underrepresented setting. The research question is well-aligned with PLOS Mental Health’s focus on global mental health equity. The study is ethically conducted, and the methodology including sampling strategy, use of validated tools, and analytical approach is generally sound. The conclusions are supported by the data presented and contribute to the evidence base needed to inform school-based mental health interventions in low- and middle-income countries (LMICs).

2. Has the statistical analysis been performed appropriately and rigorously?

Response: Yes

The statistical analyses are appropriate for the study design and objectives. The use of bi-variate and multivariable logistic regression is suitable for exploring predictors of depressive symptoms. However, the manuscript would benefit from consistent terminology (e.g., “multivariate” vs. “multiple” regression) and clearer reporting of odds ratios, confidence intervals, and cut-off points used for depression screening (particularly in the abstract and results sections). These are minor issues that can be addressed during revision and do not detract from the overall rigor of the analysis.

3. Have the authors made all data underlying the findings in their manuscript fully available?

Response: Yes

Based on the Data Availability Statement and provided documentation, the authors have made the data underlying their findings fully available following PLOS data policy.

4. Is the manuscript presented in an intelligible fashion and written in standard English?

Response: Yes

The manuscript is well-written and understandable. The English used is standard and sufficiently clear for a wide academic readership. However, there are minor issues with grammar, phrasing, and consistent use of terminology (for example, alternating between “depression,” “probable depression,” and “depressive symptoms”) that should be addressed during revision. Improving structural clarity, particularly in the Introduction and Discussion sections, will further enhance the manuscript’s readability and scholarly tone.

Additional comments for the authors-

Dear Authors,

Thank you for submitting your manuscript titled “Prevalence and predictors of depressive symptoms among school-going adolescents in Kenya.” This work provides a timely and important contribution to the field of adolescent mental health, particularly in the underrepresented rural Kenyan context. The study demonstrates methodological rigor, the use of validated tools, and a well-structured analytical framework. The focus on psychosocial risk factors such as trauma, substance use, and protective influences like adult support is particularly commendable.

However, to enhance the clarity, consistency, and academic rigor of your manuscript, I respectfully offer the following comments for your consideration:

Abstract

•Line 36: Please specify the age range of the study participants.

•Line 40: Mention the cutoff score used for PHQ-A to define probable depression.

•Line 45: Clarify whether the regression model is multivariate or multiple. Additionally, clarify whether the authors examined correlation or association.

•In the Results section, add frequency counts (n) alongside percentages where appropriate.

•Line 48: Provide the full age range (minimum and maximum values).

•Line 48: Rephrase the sentence “The prevalence of depressive symptoms ranged from 2.9% for moderately severe depression to 36.9% for mild depression” for greater clarity and readability.

Introduction

•The flow transitions abruptly from global data to Kenya-specific context. Consider restructuring into clearly defined subsections:

-Global burden and trends

-Regional context (LMICs/Sub-Saharan Africa)

-Kenya-specific findings

-Psychosocial risk factors

-Impact of COVID-19

-Research gaps and rationale

•Present the rationale and objective in a clear, standalone paragraph.

•Adopt active voice wherever possible for improved clarity and reader engagement.

•Line 96: Clarify how school-related and family-level factors interact to influence adolescent depression.

•Line 124: Include examples of other countries (in addition to Kenya) where similar research has been conducted, with appropriate citations.

Methodology

•Line 142: Condense the geographic description of Kakamega to focus on its relevance to the study context.

•Lines 154–155: There is inconsistency between the population described in the abstract/introduction (rural and semirural) and this section (urban and peri-urban). Please clarify and use consistent terminology.

•Lines 159–161: Ensure consistency in referencing the study setting. Butere subcounty is mentioned in the Study Area but not the Study Setting section; this needs to be reconciled.

•Line 166: The total sample size is not specified here though it appears later in the Results. Please state the number of participants explicitly in this section.

•Line 168: Provide the parameter values used in sample size calculation, along with appropriate reference.

•Line 188: Clearly define "probable depression." Also, note that PHQ-A is a screening tool, not diagnostic.

•Line 191: Provide clearer descriptions of all instruments used, including the cut-off scores for identifying depressive symptoms (PHQ-A).

•Line 224: Clarify whether the regression model is multivariate logistic regression or multiple logistic regression.

Results

•Include frequencies (n) along with percentages throughout the text and tables where appropriate.

•Line 236: Specify the age range of participants.

•In Table 1:

-Rename variables for clarity.

-A p-value is mentioned in the caption but does not appear in the table; please correct this inconsistency.

•Line 263: The sentence does not accurately reflect the data presented in Table 2. For instance, Table 2 shows 15.8% probable depression in Malava sub-county.

•Lines 262–264: Please specify the cutoff score used to define "probable depression" based on the PHQ-A.

•Lines 265–268: The odds ratio for urban population is provided, but the terminology ("urban," "semi-rural") is used inconsistently throughout the manuscript. Standardize terminology for clarity.

•Line 275: Clarify whether "Form One student" is being referred to. As written, the sentence implies that only one student had the lowest rate of depression.

•Tables 2–5: Please revise "95CI" to "95% CI" throughout.

Discussion

•Lines 455–462: Include appropriate citations for all claims made.

•The discussion is dense and at times lacks a thematic structure. Consider:

-Synthesizing findings under key themes

-Reducing redundant or loosely connected points

-Enhancing flow with stronger transitions

•Some statements are speculative; ensure these are supported with evidence or rephrase more cautiously.

•Inconsistencies in terminology - “depression,” “probable depression,” “depressive symptoms” are noted. It is recommended to standardize to “probable depression” throughout for consistency and accuracy.

Your study offers critical insights that can inform school-based mental health interventions in resource-constrained settings. I commend your efforts and hope these suggestions assist in further strengthening the manuscript's clarity, coherence, and scholarly value.

Reviewer #3: The study is timely and highly relevant, particularly in light of the COVID-19 pandemic, which has significantly increased mental health vulnerabilities among adolescents worldwide. Investigating depression and its associated risk factors in secondary school students within a low-resource setting such as Western Kenya offers valuable insights with both local and broader implications. The methodology section is central to evaluating the strength and credibility of the study’s findings.

Below are several suggestions aimed at enhancing the methodological clarity, rigor, and overall presentation of the research.

1. Justification for Target Age Group

The study focuses on adolescents attending secondary schools aged 13–18 years. However, it would be helpful to clarify the specific rationale for choosing this age group. Are there developmental, educational, or psychosocial factors that make this population particularly relevant for examining depression outcomes in the post-COVID-19 context?

2. Lack of a Stated Hypothesis

The manuscript does not clearly articulate a hypothesis. Presenting a specific, testable hypothesis would strengthen the study’s framing and help readers better understand the direction and expectations of the analysis.

3. Overly Detailed Geographic Description

While the description of sub-counties and school types is informative, it is somewhat lengthy and may overwhelm readers unfamiliar with the region. Summarizing this information using a table or geographic map would enhance clarity and allow readers to more easily grasp the sampling framework.

4. Missing Sampling Details

Although the methodology outlines proportional sampling by gender and form level, it does not specify the final sample size, response rate, or the handling of non-responses. These are critical for evaluating the representativeness of the sample and potential selection bias. Including these details would improve the transparency and rigor of the methodology.

5. Insufficient Information on Researcher-Designed Questionnaire

The study employs a researcher-designed questionnaire to assess additional psychosocial risk factors (e.g., bullying, family structure), yet no information is provided on its development, piloting, or psychometric properties. A brief description of the design process and any reliability testing would strengthen the credibility of the instrument.

6. Lack of Graphical Presentation in Results

The results section relies solely on tables to present findings. While tables are useful for displaying detailed numerical data, the absence of visual aids limits the interpretability and impact of the results. Given the number of variables explored—such as gender, school type, trauma, and substance use—incorporating visual elements like box plots (to show depression score distributions across subgroups), bar charts, or forest plots (for logistic regression outcomes) would enhance reader comprehension and engagement.

Reviewer #4: See the attached file for the correctly formatted comments

Peer Review

“Prevalence and Predictors of Depressive Symptoms among School-Going Adolescents in Kenya“, Magige et al.

Major Concerns

- The reliance on stepwise regression may inflate type I errors (false positives). The results of the regression could be contrasted with those of a cross-validation or penalization regression (e.g., LASSO) for variable selection.

- Many variables were included, which results in a more comprehensive evaluation of depression and associated symptoms and demographic factors. However, the introduction of so many variables may also introduce many correlated factors (which may over-adjust the model or introduce bias). There is no mention of a check for multicollinearity (e.g., through the VIF)

o Similarly, there is no mention of model discrimination (e.g., AUC/ROC). Without this, there is not assessment of model performance, i.e., how well the model fits the data or how reliably it “predicts” probable depression.

- Once again, the risk of false positives is increased by using variables in multiple bivariate analyses. Was any multiple testing correction, e.g., Bonferroni, utilized? If not, it should be explained why.

- Were any sensitivity analyses conducted? For example, control analyses with alternative cut-off points or interaction terms (e.g., known interaction between age and sex, and since there were significant age effects, but apparently non-significant sex differences). Similarly, a model that only included the variables that were statistically retained by the step-wise regression (without including the ones that didn’t seem to be relevant and were “forced” into the model, e.g., the “key plausible factors”, line 220) would be interesting as a control analysis.

Minor Concerns

- Cultural validation of PHQ? Also, effect of cultural background on symptom reporting? (It is discussed that these topics are somewhat tabu, but there are so in many countries; further elaborating this may be beneficial to understanding)

- Some elaboration on concepts that may not be internationally understood, e.g.:

o Lines 237-238: “forms two to three”; what ages are these, exactly? School systems across countries vary greatly, and although the age range for this study is adolescence, it would be helpful to know what age(s) correspond to which forms. Also, why are there students above 18? Is this common in Kenya (i.e., is schooling 13 years long instead of the 12 years that are typical in western countries, or is it because more children repeat grades or enter school later than 6y.o.)? Also in line 238, what ages were the four students that represent the smallest proportion (and how can 4 students (out of 1000) be 19%)?

o “Harambee” in Table 1 � would benefit from an explanation in the Table footing

o Line 404; “national examinations” (is it an exam taken at the end of highschool? A short subclause within the sentence to clarify this would help)

- Table 1 would profit from some formatting; also, n=1020 is the number of variables? Or of participants? The way it’s presented here is misleading. Further, it should be indicated that the numbers in brackets (starting at “County”) are percentages.

- Line 264 is missing the cut-off value, which is essential for understanding the severity of the mentioned prevalences. Specify how probable depression was defined from PHQ-A scores.

- Lines 275-277; is this a longitudinal study? How come one student was measured in the four forms? Or does this mean that the lowest prevalence per form was 9.8, 13.9, 20, 20.7%? This needs clarification because the sentence makes it seem as if it were a longitudinal study.

- Line 294; there’s a part of the sentence missing here.

- Line 341; the sentence says “after adjustment” in the beginning, but “before and after adjustment” at the end. This seems like a contradiction, which is it?

- Was there any missing data? How was this handled?

Recommendation

- Many errors are present in the text, e.g., omissions of words or whole subclauses of sentences. The text must be revisited extensively, ensuring completeness and correctness. This lack of attention to detail in the text may indicate lack of thoroughness when performing the statistical analyses.

- Many sentences and statements must be clarified, as they are not understandable or seem misleading, see comments under “Minor Concerns” above.

- Albeit the statistical methods seem thorough for a prevalence study, some additional tests may enhance the robustness and thus credibility of the results. See the recommendations under “Major Concerns”. My fear is that the effect sizes are not stable, or confounded by the logistic regression model (since we don’t know what the model performance/quality is.

- After these revisions, I recommend the paper to be accepted for submission, as it strives to fill an important gap in research regarding mental health in LMICs, which may motivate further research into culturally-dependent intervention strategies.

6. PLOS authors have the option to publish the peer review history of their article (what does this mean? ). If published, this will include your full peer review and any attached files.

**Do you want your identity to be public for this peer review?** For information about this choice, including consent withdrawal, please see our Privacy Policy .

Reviewer #1: **Yes: ** Dr David Onchonga

Reviewer #2: **Yes: ** Abir Dutta

Reviewer #3: No

Reviewer #4: No

---

## [Editor Report · Decision Letter 1]

8 Oct 2025

Prevalence and Predictors of Depressive Symptoms among School-Going Adolescents in Kenya.

PMEN-D-25-00266R1

Dear Ms Magige,

We are pleased to inform you that your manuscript 'Prevalence and Predictors of Depressive Symptoms among School-Going Adolescents in Kenya.' has been provisionally accepted for publication in PLOS Mental Health.

Best regards,

Vitalii Klymchuk, Ph.D., D.Sc.

Academic Editor

PLOS Mental Health